# Apolipoprotein Mimetic Peptides: An Emerging Therapy against Diabetic Inflammation and Dyslipidemia

**DOI:** 10.3390/biom11050627

**Published:** 2021-04-23

**Authors:** Paul Wolkowicz, C. Roger White, G. M. Anantharamaiah

**Affiliations:** Department of Medicine, University of Alabama at Birmingham (UAB), Birmingham, AL 35294, USA; wolk@uab.edu (P.W.); rogerwhite@uabmc.edu (C.R.W.)

**Keywords:** atherosclerosis, dyslipidemia, hyperglycemia, hypertriglyceridemia, high-density lipoproteins, lipoproteins, amphipathic helical peptides

## Abstract

Obesity has achieved epidemic status in the United States, resulting in an increase in type 2 diabetes mellitus, dyslipidemia, and cardiovascular disease. Numerous studies have shown that inflammation plays a key role in the development of insulin resistance and diabetic complications. HDL cholesterol levels are inversely associated with coronary heart disease in humans. The beneficial effect of HDL is due, in part, to apolipoproteins A-I and E, which possess anti-inflammatory properties. The functional quality of HDL, however, may be reduced in the context of diabetes. Thus, raising levels of functional HDL is an important target for reducing inflammation and diabetic complications. Apo A-I possesses eight alpha-helical sequences, most of which form class A amphipathic helical structures. Peptides belonging to this class inhibit atherogenesis in several mouse models. Additional peptides based on structural components of apoE have been shown to mediate a rapid clearance of atherogenic lipoproteins in dyslipidemic mice. In this review, we discuss the efficacy of apolipoprotein mimetic peptides in improving lipoprotein function, reducing inflammation, and reversing insulin resistance and cardiometabolic disease processes in diabetic animals.

## 1. Lipoprotein Abnormalities in Patients with Diabetes

Vascular complications, including coronary artery disease (CAD), are common causes of death in humans in Western countries. In patients with insulin resistance and type 2 diabetes, plasma lipid and lipoprotein abnormalities are common [1]. This includes reduced high-density lipoproteins (HDL) cholesterol levels, elevated triglyceride lipoprotein (very-low-density lipoprotein, VLDL) levels and the presence of small dense low-density lipoprotein (LDL) particles [2]. Therefore, diabetes patients have increased risk for CAD. Increased levels of VLDL are due to the enhanced secretion of large triglyceride-rich VLDL by hepatocytes and impaired clearance of VLDL. Small dense LDL particles are associated with increased CAD progression [3]. Treatment regimens for dyslipidemia include behavioral modifications (diet, exercise) and pharmacological interventions [2]. Improvement in CAD has been achieved by pharmacological treatments. Medications such as statins, fibrates, niacin, and thiazolidinediones have been used to treat lipid and lipoprotein abnormalities associated with insulin resistance and type 2 diabetes, resulting in significant improvement in coronary artery disease after diabetic dyslipidemia treatment [2,3].

HDL cholesterol has been shown to be inversely correlated with cardiovascular disease risk [3]. HDL participates in the hepatic clearance of excess cellular cholesterol. This process is called reverse cholesterol transport (RCT). HDL also possesses anti-inflammatory properties [4]. These include prevention of LDL oxidation, which inhibits monocyte chemotactic activity in endothelial and human aortic smooth muscle cells. HDL also attenuates the inflammatory response to tumor necrosis factor-1 (TNF-1) and interleukin-1 (IL-1), and possesses antithrombotic and antiapoptotic properties [5,6]. Recent studies have shown that HDL function, rather than HDL levels, are important for the protective effects of HDL [4,7]. Increasing plasma HDL cholesterol levels has been proposed as a therapeutic option for reducing CAD risk in type 2 diabetic patients [5,6]. Data suggest that a reduction in HDL may impair pancreatic β-cell insulin secretion and skeletal-muscle glucose uptake [8]. Genetic variants of HDL are not always associated with increased risk for type 2 diabetes [9]. Recently, the claim that HDL function, rather than HDL levels, is important for protecting against cardiovascular complications is gaining acceptance [10,11,12].

## 2. Modification of Apolipoprotein (apo) A-I in Type 2 Diabetes Patients and Consequences

In diabetes patients, HDL becomes dysfunctional due to various factors [13]. HDL proteins can be modified via glycation and/or oxidation. Further, due to elevated circulating oxidized lipids, the activities of HDL-associated enzymes such lecithin:cholesterol acyltransferase (LCAT), paraoxonase 1 (PON-1), and lipoprotein-associated phospholipase A2 (PLA_2_) are altered [14,15]. A reduction in the activities of these antioxidant enzymes reduces HDL function, which is involved in vascular protection from oxidative damage [15,16,17]. Hyperglycemia is a common feature in diabetic patients and has been shown to cause glycation of apoA-I [18], which alters its structure and function. For example, it has been shown that glycation of apoA-I has a direct effect on the LCAT-mediated conversion of plasma cholesterol into cholesteryl ester [19]. Glycation of apoA-I in HDL also diminishes clearance of cholesterol from the peripheral tissues in both type 1 and type 2 diabetic subjects [20,21,22].

Glycation of apoA-I has been shown to be associated with plaque progression in patients with CAD [23]. Under normal physiological conditions, apoA-I stimulates AMP-activated protein kinase and improves glucose metabolism [24]. Advanced glycation of apoA-I, however, exerts adverse effects on levels of circulating glucose. Several specific Lys residues in apoA-I (K12, K23, K40, K96, K106, K107, and K238) have been shown to undergo glycation. This results in the impairment of anti-inflammatory function in type 2 diabetics, including an impaired inhibition of LPS-mediated TNFα and IL1β secretion [25]. In these studies, glycation of apoA-I was shown to be inversely correlated with its anti-inflammatory function [26].

There are 22 Lys residues in the apoA-I sequence (Figure 1). ApoA-I consists of 22-mer tandem-repeating lipid-associating domains, most of them punctuated by a Pro residue [27]. Studies in our laboratory showed that among eight 22 mer domains, only the end two domains (peptides corresponding to 44–65 and 220–243 region of apoA-I primary sequence) are able to associate with lipids spontaneously [28]. Among the Lys residues susceptible to glycation (Figure 1), all of the Lys residues are at the *N*-terminal end, with one exception at the C-terminal (K238) of the protein. In the amphipathic helical structure, these Lys residues are at the polar–nonpolar interface. When apoA-I is associated with lipid, as present in HDL, the protein is stabilized by the salt bridges with the Asp and Glu residues from the adjacent amphipathic helical strand, as suggested by the double-belt structure of apoA-I on either discoidal or spherical HDL particles [29]. Therefore, it is tempting to hypothesize that glycation of Lys residues at the lipid-associating sites of apoA-I causes the protein to disassociate from the lipid surface, making the protein more susceptible for proteolysis. This process decreases the levels of circulating HDL in diabetes patients.

## 3. Apolipoprotein Mimetic Peptides in the Treatment of Diabetes

Modes of treatment of diabetic dyslipidemia include clearance of atherogenic lipoproteins to correct dyslipidemia and inhibition of oxidative stress due to increased lipid hydroperoxides [30]. While HDL and apoA-I supplementation may correct the diabetic complications due to low levels of HDL, isolation of large quantities of HDL and apoA-I from human plasma that are functionally active is not practical. Therefore, there is a need to design apoA-I mimetics, or mimics of apoE (the protein in VLDL responsible for clearance of large amounts of plasma cholesterol), that are small in size and amenable to organic synthesis in large quantities.

The first breakthrough took place in our laboratory when we were studying mechanisms by which apolipoproteins associate with lipids [31]. As mentioned earlier, apoA-I possesses tandem-repeating amphipathic helical domains. The general structure of amphipathic helical domains in apolipoproteins consists of an α-helical structure with opposing polar–nonpolar faces. Specifically, positively-charged amino acids are arranged at the polar–nonpolar interface, and negatively-charged residues are at the center of the polar face [32]. To support the amphipathic helix theory as the mechanism of lipid association of apolipoproteins, we designed an 18-residue peptide that does not possess sequence homology with any of the exchangeable apolipoproteins, but when folded as an α-helix, forms opposing polar–nonpolar faces with Lys residues at the polar interface and Asp and Glu at the center of the polar face [31]. This peptide, called 18A, spontaneously associates with multilamellar vesicles of dimyristoyl phosphatidylcholine (DMPC) to form a discoidal particle whose size and shape were similar to those formed by apoA-I and DMPC. Several years later, an analog of this baseline peptide called 5F was administered to C57BL6 mice on an atherogenic diet in order to study the effect of the peptide on the lesion formation [33]. We observed that the 5F peptide inhibited lesion formation in mice administered a high-fat diet by a mechanism that does not involve changes in the lipoprotein profile. Furthermore, when HDL was isolated from mice administered a high-fat diet injected with 5F and presented to human artery wall endothelial cells in vitro together with human low-density lipoprotein (LDL), there were substantially fewer lipid hydroperoxides formed. This resulted in substantially lower LDL-induced monocyte chemotactic activity than with HDL from PBS-injected animals. Thus, the peptide with only 18 amino acid residues was shown to possess anti-inflammatory properties [33].

Based on these observations, Abraham and coworkers used the 4F analog of apoA-I mimetic peptide to study effects of peptide administration on hyperglycemia, obesity, and metabolic syndrome using animal models of diabetes [34,35]. In the first experiment performed using peptide L-4F, they studied whether the peptide induces antioxidative enzymes that are vasoprotective in a rat model of diabetes. They also explored whether L-4F ameliorates insulin resistance and diabetes in obese mice [34]. The 4F treatment increased aortic and bone marrow heme oxygenase (HO-1) activity and decreased aortic and bone marrow superoxide production (*p* < 0.001) [36]. The 4F treatment also increased serum adiponectin levels, and decreased adipogenesis in mouse bone marrow and in cultures of human bone marrow-derived mesenchymal stem cells [37]. This resulted in reduced adiposity, improved insulin sensitivity, glucose tolerance, and increased plasma adiponectin levels [34,37]. Peptide 4F administration reduced inflammatory markers such as IL-1beta and IL-6 in obese mice [37]. Peptide L-4F was administered in obese mice to study the effect of the peptide on antioxidant enzymes such as HO-1. Vascular function was studied by measuring the expression of pAMPK, pAKT, and insulin-receptor phosphorylation [35]. L-4F was injected intraperitoneally at a dose of 200 µg/100 g daily for six weeks into *ob/ob* mice. The treatment resulted in a significant reduction in adiposity. There was a decrease in visceral and subcutaneous fat levels and a reduction in adipogenesis. They also observed a smaller increase in insulin-sensitive adipocytes. This response was due to an increase in pAMPK, pAKT, and insulin-receptor phosphorylation [35].

The effect of D-4F administration has also been studied in acute hyperglycemia [38]. It was shown that the peptide protects against the attenuation of eNOS activity induced by acute hyperglycemia. The 4F peptide reduces superoxide generation and thus restores eNOS signaling and NO biosynthesis in acute hyperglycemia [39]. Peptide 4F has been shown to reduce plaque inflammation and prevent atherosclerosis in a diabetic apoE null mouse model [39]. These studies demonstrated that 4F was effective in preventing accelerated atherosclerosis in a mouse model of pre-existing diabetes. These effects were associated with a significant reduction of hepatic arachidonic acid and oxidized fatty-acid levels. Krugher et al. showed that administration of D-4F to Sprague–Dawley rats also administered streptozotocin (STZ) induced increased levels and activity of HO-1 and superoxide dismutase (SOD). In addition, there was a significant decrease in superoxide anion levels [39]. In HO-2 knockout mice, administration of 4F restored HO-1 levels and activity, and increased adiponectin levels [40]. In all of the studies performed with apoA-I mimetic peptides, plasma cholesterol was not reduced, and no changes in the levels of atherogenic lipoproteins or cholesterol profiles were observed, despite beneficial effects on the diabetes-mediated oxidative stress.

## 4. Effect of ApoE Mimetic Peptides on Plasma Cholesterol and Diabetes in Mouse Models of Diabetes

ApoE, the protein component in VLDL, is a second ligand for the clearance of LDL and, more importantly, is involved in the clearance of VLDL and chylomicrons [41]. It binds to a number of alternate receptors and thus clears atherogenic lipoproteins via the non-LDL receptor pathway [42]. In addition to atherogenic cholesterol-lowering properties, apoE possesses many cholesterol-independent properties [43]. ApoE is therefore a good candidate for the treatment of vascular complications associated with hyperglycemia. However, apoE is a large protein consisting of 299 amino acids and is not practical to either isolate or produce large quantities by genetic-engineering technologies. The apoE sequence possesses a two-domain structure consisting of a long lipid-associating domain at the C-terminus (residues 202–263 in apoE) and a receptor-binding domain (residues 141–150) at the *N*-terminal end [44]. Since we had found that 18A can associate with lipids, we designed a peptide in which the receptor-binding domain (LRKLRKRLLR, 141–150 residues from apoE) was covalently linked to 18A. A peptide containing 28 amino acid residues (Ac-LRKLRKRLLR-DWLKAFYDKVAEKLKEAF-NH_2_) was synthesized and was shown to enhance cellular uptake of atherogenic lipoproteins via the HSPG pathway [45]. This apoE mimetic peptide (Ac-hE18A-NH_2_) has also been shown to dramatically reduce plasma cholesterol and inhibit atherosclerosis [46]. It also inhibits inflammation by promoting nitric oxide formation, enhancing PON-1 activity, and improving the clearance of lipid hydroperoxides in several dyslipidemic animal models [47,48,49,50]. Administration of an apoE mimetic peptide into dyslipidemic C57BL6/J mice increased plasma insulin levels, improved insulin sensitivity, reduced plasma glucose levels, and decreased body weight (Figure 2) [50]. Table 1 summarizes the effect of apolipoprotein mimetic peptides in animal models of diabetes.

Recently, our laboratory has shown that the acylation of the *N*-terminal of Ac-hE18A-NH_2_ enhanced the ability of the peptide to clear plasma cholesterol in dyslipidemic monkey models [51]. An apoE mimetic peptide analog has recently been shown to attenuate cellular injury in LPS-treated THP-1 macrophages and facilitate the removal of cellular debris and damaged organelles via induction of autophagy. In particular, the peptide was shown to reduce mitochondrial superoxide formation, prevent the LPS-induced decrease in mitochondrial membrane potential, and attenuate the release of cytochrome *c*. The modified peptide also inhibited the activities of initiator caspases 8 and 9 and effector caspase 3. The attenuation of apoptosis in AEM-2-treated cells was associated with an increase in cellular autophagy [52]. Based on these exciting results, apoE mimetic peptides may be ideal candidates for the treatment of inflammatory complications associated with severe hypertriglyceridemia and diabetes.

## 5. Conclusions

Despite recent developments in the treatment of diabetes, there remains a strong need to identify new therapies to ameliorate the vascular complications associated with dyslipidemia, inflammation, and cardiovascular disease. The apolipoproteins apoA-I and apo-E present in HDL are anti-inflammatory and correct serum dyslipidemias. Based on these observations, peptides derived from the primary sequences of apoA-I and apoE have been synthesized that mimic their vasculo-protective properties. In animal models of diabetes, these A-I and E peptides reduce serum lipid hydroperoxides, increase anti-oxidant proteins present in HDL, and decrease serum atherogenic lipoproteins. Peptides containing 18 to 28 amino acids can be synthesized in large quantities compared to the 243-amino-acid apolipoprotein A-I and the 299-amino-acid Apo E; both must be isolated from human plasma to be used as therapeutic agents, a daunting task. Further studies will determine if these peptides are effective therapeutic interventions that stanch the vascular complications that accompany diabetes.

## Figures and Tables

**Figure 1 biomolecules-11-00627-f001:**
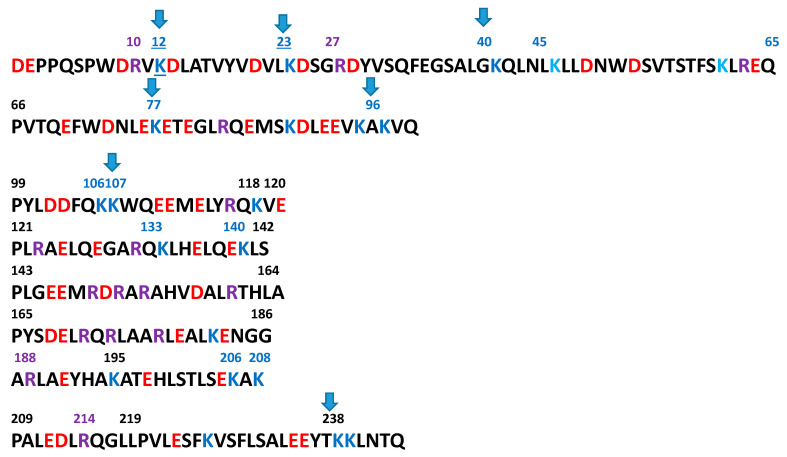
Sequence of human apoliprotein A-I. Points of glycation of Lys residues in the apoA-I sequence are shown by the arrows. Glycation of apoA-I perturbs the lipid-associating property of apoA-I, thus rendering HDL unstable. Acidic amino acids are in red. Lys residues are in blue and Arg residues in purple. Arrow marks represent the points of glycation of Lys residues.

**Figure 2 biomolecules-11-00627-f002:**
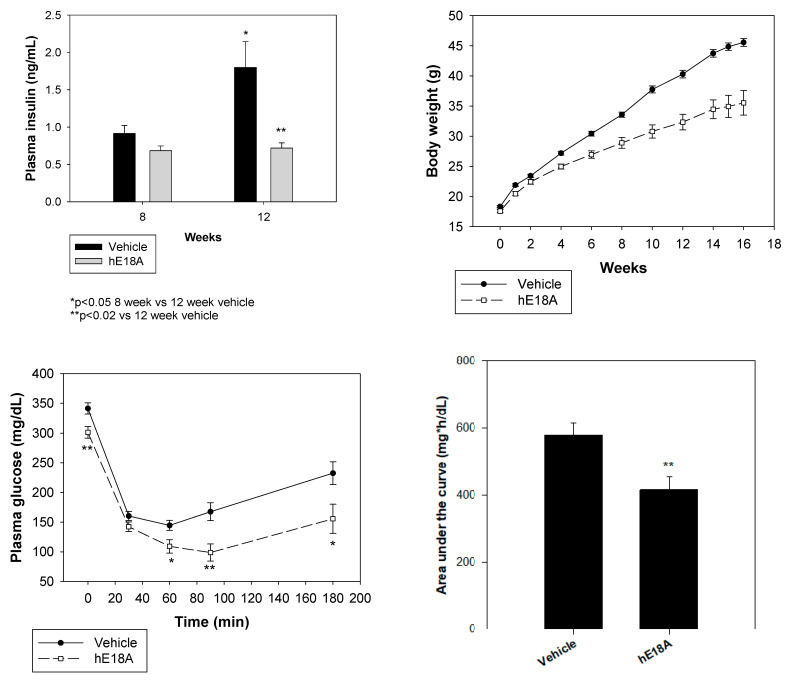
Effect of apoE mimetic peptide Ac-hE18A-NH_2_ on insulin resistance. Male C57BL6/J mice were fed a diet containing 60% fat by calories at five weeks of age, and treatment was immediately begun with vehicle or AEM-28 (three times weekly i.v., 200 μg/mouse) and continued through 18 weeks (*N* = 15 in each group). (**Top Left**) Plasma insulin levels increased in control mice but not in peptide-administered mice. (**Top Right**) Body weight decreased significantly in peptide-administered mice. (**Bottom Left**) An insulin-tolerance test showed improvement in insulin resistance, as shown by a significant decrease in plasma glucose levels. (**Bottom Right**) Area under the curve from panel C shows a significant decrease in glucose levels. * *p* < 0.05 vs vehicle; ** *p* < 0.02 vs vehicle.

**Table 1 biomolecules-11-00627-t001:** Summary of effects of apolipoprotein mimetic peptides in diabetes animal models.

Peptide	Animal Model	Biological Changes	References
4F	Obese mice	Adiposity ↓	[34]
		Bone marrow adipogenesis ↓	[34]
		Insulin sensitivity ↑	[34,35]
		Glucose tolerance ↑	[35,37]
		IL-b, IL-6, superoxide ↓	[37]
		Hepatic lipid content ↓	[35,37]
		Adipocytes of small size ↑	[35,37]
D-4F	Sprague–Dawley Rats with STZ	HO-1 ↑ SOD ↑	[36]
		Endothelial sloughing ↓	[36]
4F	HO-2 knockout mice	HO-1 ↑ Adeponectin ↑	[40]
Ac-hE18A-NH_2_	C57B6/fat administered	Plasma insulin ↑	[50]
		Insulin sensitivity ↑	
		Plasma glucose ↓	

## Data Availability

Not applicable.

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
