# Peer review of "Apolipoprotein Mimetic Peptides: An Emerging Therapy against Diabetic Inflammation and Dyslipidemia"

_biomolecules, 2021, doi:10.3390/biom11050627_

Round 1

Reviewer 1 Report

The proposed review reports on the ability of the apolipoprotein mimetic peptides to improve lipoprotein function, reduce inflammation and reverse insulin resistance and cardiometabolic disease processes in diabetic animals.

It is a short and ot informativereview, needs an extensive revision and professional language editing.

The review containing long sentences written in a vaguely way with repetitive words at the same sentence that make it unclear and difficult for reading and understanding.

 For example see paragraphs  Lines 47-48; 77-89; 104-114; 123-130 and 177-188.

Line 26 : "CAD is the most common cause of death": 

change to human death, and add where is it common? 

Line 54 : protecting againstcardiovascular please correct to “against cardiovascular”

Line 66: “clearance of cholesterol ester via the SRB1 pathway” 

HDL is responsible for the clearance of free cholesterol via SRB1, which is then becomes cholesteryl ester via LCAT.

Line 118 “human artery wall cells “    add which cells (endothelial or SMC)?

I would suggest adding a table that summarize the results obtained from the different peptides injected to different mice models.

  I would suggest adding a comprehensive conclusion.

Author Response

Reviewer 1.

The proposed review reports on the ability of the apolipoprotein mimetic peptides to improve lipoprotein function, reduce inflammation and reverse insulin resistance and cardiometabolic disease processes in diabetic animals.

It is a short and ot informativereview, needs an extensive revision and professional language editing.

The review containing long sentences written in a vaguely way with repetitive words at the same sentence that make it unclear and difficult for reading and understanding.

 For example see paragraphs  Lines 47-48; 77-89; 104-114; 123-130 and 177-188.

Line 26 : "CAD is the most common cause of death": 

change to human death, and add where is it common? 

Line 54 : protecting againstcardiovascular please correct to “against cardiovascular”

Line 66: “clearance of cholesterol ester via the SRB1 pathway” 

HDL is responsible for the clearance of free cholesterol via SRB1, which is then becomes cholesteryl ester via LCAT.

Line 118 “human artery wall cells “    add which cells (endothelial or SMC)?

I would suggest adding a table that summarize the results obtained from the different peptides injected to different mice models.

I would suggest adding a comprehensive conclusion.

We would like to thank you for reviewing the article biomolecules-1144447 entitled “ Apolipoprotein mimetic peptides in vascular protective effects, dysfunction and potential as therapeutic target for diabetes”.
We have carefully considered all comments and prepared a revised version of the manuscript in accordance with guidelines.

All of the long run-on sentences have been corrected as per reviewer's suggestions.

Grammatical errors have been corrected.

A New table has been incorporated as suggested by the reviewer.

Line 54 has been corrected

Line 66 has been rewritten to make it clear.

Line 118 has been modified to include the type of cell line.

A table has been incorporated.

Reviewer 2 Report

The authors have given a succinct overview of the potential of apolipoprotein mimetic peptides in combatting vascular comorbidities associated with diabetes. Specifically, the authors note the relationship between high density lipoproteins and coronary artery disease, establishing the logic that maintenance, or indeed augmenting high density lipoproteins, may have beneficial effects in a vascular disease context. In turn, the argument for utilising apolipoprotein, or mimetics of such, is put forward by the authors, with a body of evidence containing new data provided within, and also supported by similar literature. Taken together, this was an interesting topic to read about.

In reviewing the manuscript, I have a number of concerns however. The following should be considered when preparing a suitable revision.

  1. The writing of the manuscript could be improved throughout. While the points being made can be interpreted, there are a number of grammatical errors which greatly disrupt the flow of the piece. At times, the information reads like bullet points than a balanced paragraph. The authors must address this in any resubmission.
  2. The style of referencing needs to be addressed. Typically, numbers are used in place of the roman numerals for ease. The authors should address this in any resubmission.
  3. Similarly, the referencing itself could be improved. A great number of the references are quite old, and could be replaced with more up-to-date counterparts, whilst the frequency/presence of referencing in some sections could be improved/is absent. There are paragraphs where there isn’t a single reference. The authors must improve on this in advance of any resubmission.
  4. In reviewing the article, I was unsure as to what type of article this was, whether it was an original research article, or a review, or something else entirely. If it is a review, the focus of the paper on the authors previous works is at times too much, and instead a more balanced overview of the research area is needed. If it is a research article owing to the inclusion of data, then information such as the methods and materials are needed to support the data, if even as a supplementary. The authors must address this point in any resubmission.

Author Response

Reviewer 2.

The authors have given a succinct overview of the potential of apolipoprotein mimetic peptides in combatting vascular comorbidities associated with diabetes. Specifically, the authors note the relationship between high density lipoproteins and coronary artery disease, establishing the logic that maintenance, or indeed augmenting high density lipoproteins, may have beneficial effects in a vascular disease context. In turn, the argument for utilising apolipoprotein, or mimetics of such, is put forward by the authors, with a body of evidence containing new data provided within, and also supported by similar literature. Taken together, this was an interesting topic to read about.

In reviewing the manuscript, I have a number of concerns however. The following should be considered when preparing a suitable revision.

  1. The writing of the manuscript could be improved throughout. While the points being made can be interpreted, there are a number of grammatical errors which greatly disrupt the flow of the piece. At times, the information reads like bullet points than a balanced paragraph. The authors must address this in any resubmission.
  2. The style of referencing needs to be addressed. Typically, numbers are used in place of the roman numerals for ease. The authors should address this in any resubmission.
  3. Similarly, the referencing itself could be improved. A great number of the references are quite old, and could be replaced with more up-to-date counterparts, whilst the frequency/presence of referencing in some sections could be improved/is absent. There are paragraphs where there isn’t a single reference. The authors must improve on this in advance of any resubmission.
  4. In reviewing the article, I was unsure as to what type of article this was, whether it was an original research article, or a review, or something else entirely. If it is a review, the focus of the paper on the authors previous works is at times too much, and instead a more balanced overview of the research area is needed. If it is a research article owing to the inclusion of data, then information such as the methods and materials are needed to support the data, if even as a supplementary. The authors must address this point in any resubmission.

We would like to thank you for reviewing the article biomolecules-1144447 entitled “ Apolipoprotein mimetic peptides in vascular protective effects, dysfunction and potential as therapeutic target for diabetes”.

We have carefully considered all comments and prepared a revised version of the manuscript in accordance with guidelines.

  1. We made sincere attempts to erase grammatical errors. Several sections have been rewritten
  2. we have followed the reference style recommended by the journal. I do not understand the reference roman numerals.
  3. Several new references have been added to statements.
  4. This is written as a mini review.
  5. I have been told by the inviting editor that there is no such review on this new line of research in the field of diabetes. Therefore, he would like to see this published in the special issue for which he is the coeditor.

Reviewer 3 Report

This is an interesting review of the current understanding and recent findings with respect to the use of anti mimetic peptides in vascular protection.

My only comment is that line 60 does not explain how LCAT and PON-1 and PLA2 are altered and why that is important? 

Author Response

Reviewer 3.

This is an interesting review of the current understanding and recent findings with respect to the use of anti mimetic peptides in vascular protection.

My only comment is that line 60 does not explain how LCAT and PON-1 and PLA2 are altered and why that is important? 

We would like to thank you for reviewing the article biomolecules-1144447 entitled “ Apolipoprotein mimetic peptides in vascular protective effects, dysfunction and potential as therapeutic target for diabetes”.

We have carefully considered all comments and prepared a revised version of the manuscript in accordance with guidelines.

The only concern he has about the importance of enzyme activities has been addressed.

Round 2

Reviewer 1 Report

The authors addressed al the points raised and improved their MS and I think that now it is suitable for publication. 

minor revisions 

1- In page 3 the authors wrote "domains, only the end
two are able to associate with lipids spontaneously"  what they mean by end two?

2- In the same page 3 the authors wrote " all of the other Lys residues are at the C-terminal end with one at the C-terminal (K238) of the protein".  Is not clear if all the Lys residues including K238 at the C-terminal  or there is a mistake and one at the N-terminal and the other at C-terminal . 

3- page 4 the authors wrote " To study effect 4F on antioxidant enzymes
such as HO-1 in obese mouse models, peptide 4F was administered in obese
mice"  may be changed to "  peptide 4F was administered in obese
mice study the effect of 4F on antioxidant enzymes such as HO-1. 

Author Response

  1. We have now made this point clear by giving the residue numbers of the two peptides that spontaneously associate with lipids.
  2. we have now re-written this sentence and made it clear.
  3. we have made the suggested change

Reviewer 2 Report

The authors have responded positively to my comments and in addressing my points the manuscript is much improved.

However, there are some points that still need attention/clarification:

  1. While the authors have made alterations to the writing of the manuscript, some sections still have grammatical errors, and some of the rewrites themselves contain grammatical errors. There are too many to list individually, but the authors should look to correct these in advance of any resubmission.
  2. My experience with articles in MDPI journals, and indeed Biomolecules, is that the referencing follows a [1][2][3]… style ahead of a [i][ii][iii]… style. That is what’s meant by using ‘roman numerals’ instead of ‘numbers’. In fact, the authors adopt a numbering strategy at times in the text, but for the entire table which is now added, which means two styles are adopted. Adopting one style over another is needed, but perhaps MDPI are satisfied with you using roman numerals - I was highlighting it as something which might improve the manuscript as numbers are easier to locate in a reference list ahead of roman numerals.
  3. The addition of new references to support some of the information provided is appreciated, and the manuscript is much improved as a result of these additions.
  4. Thank you for clarifying that this is a mini review. At the same time, as there is experimental data, I would request the authors include methodology that compliments these data, or else include references in which similar experiments were conducted/these experiments are based upon.

Author Response

  1. We have carefully re-edited the manuscript and corrected mistakes.  We have also changed the title of the manuscript to better suit the body of the review. 
  2. We thank the editors for making the required change in the referencing format.
  3. We have included the experimental details in the figure legend.